# Universal third-trimester ultrasonic screening using fetal macrosomia in the prediction of adverse perinatal outcome: A systematic review and meta-analysis of diagnostic test accuracy

Alexandros A. Moraitis[1], Norman Shreeve[1], Ulla Sovio[1], Peter Brocklehurst[2], Alexander E. P. Heazell[3,4], Jim G. Thornton[5], Stephen C. Robson[6], Aris Papageorghiou[7], Gordon C. Smith[1]*

1 Department of Obstetrics and Gynaecology, University of Cambridge, NIHR Cambridge Comprehensive Biomedical Research Centre, Cambridge, United Kingdom, 2 Birmingham Clinical Trials Unit, University of Birmingham, Birmingham, United Kingdom, 3 Maternal and Fetal Health Research Centre, School of Medical Sciences, Faculty of Biological, Medical and Health, University of Manchester, Manchester Academic Health Science Centre, Manchester, United Kingdom, 4 St. Mary's Hospital, Central Manchester University Hospitals NHS Foundation Trust, Manchester Academic Health Science Centre, Manchester, United Kingdom, 5 Division of Child Health, Obstetrics and Gynaecology, School of Medicine, University of Nottingham, Nottingham, United Kingdom, 6 Reproductive and Vascular Biology Group, The Medical School, University of Newcastle, Newcastle, United Kingdom, 7 Nuffield Department of Obstetrics and Gynaecology, Oxford, United Kingdom

* gcss2@cam.ac.uk

**Data Availability Statement:** All relevant data are within the manuscript and its Supporting

## Abstract

### Background

The effectiveness of screening for macrosomia is not well established. One of the critical elements of an effective screening program is the diagnostic accuracy of a test at predicting the condition. The objective of this study is to investigate the diagnostic effectiveness of universal ultrasonic fetal biometry in predicting the delivery of a macrosomic infant, shoulder dystocia, and associated neonatal morbidity in low- and mixed-risk populations.

### Methods and findings

We conducted a predefined literature search in Medline, Excerpta Medica database (EMBASE), the Cochrane library and ClinicalTrials.gov from inception to May 2020. No language restrictions were applied. We included studies where the ultrasound was performed as part of universal screening and those that included low- and mixed-risk pregnancies and excluded studies confined to high risk pregnancies. We used the estimated fetal weight (EFW) (multiple formulas and thresholds) and the abdominal circumference (AC) to define suspected large for gestational age (LGA). Adverse perinatal outcomes included macrosomia (multiple thresholds), shoulder dystocia, and other markers of neonatal morbidity. The risk of bias was assessed using the Quality Assessment of Diagnostic Accuracy Studies (QUADAS-2) tool. Meta-analysis was carried out using the hierarchical summary receiver

Information files. All the studies included in the meta-analysis are publicly available.

**Funding:** This study was funded by the National Institute for Health Research (NIHR) Health Technology Assessment programme, grant number 15/105/01. The funders had no role in study design, data collection and analysis, decision to publish, or preparation of the manuscript.

**Competing interests:** I have read the journal's policy and the authors of this manuscript have the following competing interests: AAM, NS, PB, JGT, and SCR have no competing interests to declare. US reports grants from NIHR Cambridge Biomedical Research Centre during the conduct of the study. GCS is a member of the Editorial Board of PLOS Medicine. GCS reports grants and personal fees from GlaxoSmithKline Research and Development, Ltd., grants from Sera Prognostics, Inc., nonfinancial support from Illumina, Inc., and personal fees from Roche Diagnostics, Ltd., outside the submitted work. In addition, GCS and US have a patent in preparation for a novel predictive test for fetal size pending. AP reports personal fees from educational events/lectures, clinical services in the private sector and from Consultancy via Oxford University Innovation, royalties from published works, and editorial work for UOG and BJOG, outside the submitted work.

**Abbreviations:** AC, abdominal circumference; BPD, Biparietal diameter; CDSR, Cochrane Database of Systematic Reviews; CENTRAL, Cochrane Central Register of Controlled Trials; CI, confidence interval; DOR, diagnostic odds ratio; EFW, estimated fetal weight; EMBASE, Excerpta Medica database; FL, femur length; HC, head circumference; HSROC, hierarchical summary receiver–operating characteristics; IOL, induction of labor; LGA, large for gestational age; LR, likelihood ratio; NICE, National Institute for Health and Care Excellence; PRISMA, Preferred Reporting Items for Systematic Reviews and Meta-Analyses; QUADAS 2, Quality Assessment of Diagnostic Accuracy Studies; RCT, randomized controlled trial; ROC, receiver operating characteristic; SGA, small for gestational age; wkGA, weeks' gestation.

operating characteristic (ROC) and the bivariate logit-normal (Reitsma) models. We identified 41 studies that met our inclusion criteria involving 112,034 patients in total. These included 11 prospective cohort studies ($N = 9986$), one randomized controlled trial (RCT) ($N = 367$), and 29 retrospective cohort studies ($N = 101,681$). The quality of the studies was variable, and only three studies blinded the ultrasound findings to the clinicians. Both EFW >4,000 g (or 90th centile for the gestational age) and AC >36 cm (or 90th centile) had >50% sensitivity for predicting macrosomia (birthweight above 4,000 g or 90th centile) at birth with positive likelihood ratios (LRs) of 8.74 (95% confidence interval [CI] 6.84–11.17) and 7.56 (95% CI 5.85–9.77), respectively. There was significant heterogeneity at predicting macrosomia, which could reflect the different study designs, the characteristics of the included populations, and differences in the formulas used. An EFW >4,000 g (or 90th centile) had 22% sensitivity at predicting shoulder dystocia with a positive likelihood ratio of 2.12 (95% CI 1.34–3.35). There was insufficient data to analyze other markers of neonatal morbidity.

## Conclusions

In this study, we found that suspected LGA is strongly predictive of the risk of delivering a large infant in low- and mixed-risk populations. However, it is only weakly (albeit statistically significantly) predictive of the risk of shoulder dystocia. There was insufficient data to analyze other markers of neonatal morbidity.

## Author summary

### Why was this study done?

- There is a debate regarding introducing universal third-trimester screening for macrosomia. An effective screening program requires two elements: an effective test at predicting a condition and an effective intervention.

- There is evidence that early-term induction of labor (IOL) could reduce the rates of shoulder dystocia. However, there is no high-quality evidence regarding the diagnostic effectiveness of fetal biometry at predicting macrosomia and associated morbidity.

### What did the researchers do and find?

- We searched more than 10,000 titles and identified 41 studies including 112,034 patients that offered third-trimester ultrasounds for the prediction of macrosomia as part of universal ultrasound screening or were done in low- and mixed-risk populations. The quality of the studies was variable, and only three studies blinded the ultrasound findings to the clinicians.

- We found that the two most common ultrasound markers, the estimated fetal weight (EFW) and the abdominal circumference (AC), could predict the majority of macrosomic infants at birth (sensitivity >50%) with high diagnostic performance (positive LRs between 7 and 10).

- However, the EFW could only predict about 1 in 5 cases of shoulder dystocia (22% sensitivity) with low diagnostic performance (positive likelihood ratio of about 2). There was insufficient data to analyze other markers of neonatal morbidity.

### What do these findings mean?

- Universal third-trimester ultrasound screening will identify more pregnancies with macrosomia. However, it will not have a clinically significant effect at predicting shoulder dystocia. There is not enough evidence on the effect of ultrasound screening on neonatal morbidity.

- We recommend caution prior to introducing universal third-trimester screening for macrosomia, as it would increase the rates of intervention, with potential iatrogenic harm, without clear evidence that it would reduce neonatal morbidity.

## Introduction

Macrosomia is usually defined as birthweight >4,000 g or >90th centile for sex and gestational age. Macrosomic birth weight is associated with the risk of adverse outcomes, including perinatal death [1] and injuries related to traumatic delivery [2]. Ultrasonic estimated fetal weight (EFW) was first described in 1975 [3]. The equation for EFW that is in most widespread use was published by Hadlock and colleagues in 1985 [4], and the distribution of EFW in relation to week of gestation was published in 1991 [5]. Hence, the diagnostic tools to identify small for gestational age (SGA) and large for gestational age (LGA) fetuses have been available for many years. One of the main complications associated with macrosomia is shoulder dystocia, and a Cochrane review of four randomized controlled trials (RCTs) including 1,190 women demonstrated that routine induction of labor (IOL) for suspected LGA may prevent this outcome [6]. However, it remains unclear whether screening and intervention for suspected LGA results in better outcomes.

An RCT of IOL in women with an ultrasonically suspected LGA infant is in progress in the United Kingdom (The Big Baby trial, ISRCTN18229892). However, the women recruited to this trial will have been scanned because they were high risk for some reason, as the National Institute for Health and Care Excellence (NICE) has recommended that women should not be routinely scanned in late pregnancy [7]. Although the trial will confirm whether IOL is effective in high-risk women, it will not determine whether screening women without risk factors and intervening results in net benefit. It is often the case that screening and intervention programs that work well in high-risk groups do not work as well in low-risk populations, and one explanation for this can be that the screening test is less informative in low- and mixed-risk populations due to the lower prior risk of disease. In this study, we sought to quantify the diagnostic effectiveness of screening for fetal macrosomia and associated complications using universal ultrasonic fetal biometry in late pregnancy.

## Methods

### Sources

The protocol for this review was prospectively written and registered with PROSPERO (the International Prospective Register of Systematic Reviews), and the registration number was

CRD42017064093. We searched the literature systematically using the Cochrane Database of Systematic Reviews (CDSR), Cochrane Central Register of Controlled Trials (CENTRAL), Medline, EMBASE, and ClinicalTrials.gov from inception to August 2019. An update search was done on May 28, 2020. We applied no restrictions on the language of the report or the location of the study. The studies were identified using a combination of words related to "ultrasound," "pregnancy," "estimated fetal weight," "EFW," "birthweight," "macrosomia," "large for gestational age," "shoulder dystocia," and "brachial plexus injury." The exact search strategy is presented in S1 Text.

## Study selection

We set out to include cohort studies where an ultrasound scan was performed ≥24 weeks' gestation (wkGA), excluding multiple pregnancies. We included studies of low-risk populations, universal screening, and mixed-risk populations (i.e., included both high-risk and low-risk pregnancies). Studies that included only high-risk women, such as patients with preexisting or gestational diabetes, and those in which the ultrasound was performed during labor were excluded. Studies were not selected on the basis of the definition of the index test, i.e., the formula and the threshold used. Finally, we included both blinded and unblinded studies.

## Index tests and outcomes

For the purposes of the meta-analysis, we defined suspected LGA as a fetus with an EFW >4,000 g or >90th centile or with an abdominal circumference (AC) >36 cm or >90th centile. However, we have also documented other thresholds used. The outcomes studied included macrosomic birth weight (>4,000 g or >90th centile) and severe macrosomic birth weight (>4,500 g or >97th centile); shoulder dystocia; and perinatal morbidity (neonatal unit admission, 5-minute Apgar score of six or less, metabolic acidosis, neonatal hypoglycaemia, and neonatal jaundice).

## Quality assessment

Two authors (AAM and NS) independently performed the literature search, using the software package Review Manager 5.3. Any differences were addressed in consultation with the senior author (GCS). The Quality Assessment of Diagnostic Accuracy Studies (QUADAS 2) tool was used to assess the risk of biases, following the *Cochrane Handbook of Diagnostic Test Accuracy Studies* [8]. The QUADAS 2 tool was employed to assess potential biases in patient selection, index test, reference standard, and flow and timing. In relation to flow and timing, we assessed the risk from the perspective of universal ultrasound screening near term (i.e., around 36 wkGA). Flow and timing are based on the timing of the ultrasound scan, the timing of delivery, and the length of the interval between scan and delivery. A standardized data extraction form was employed to obtain information on the characteristics of the study (publication year, location, setting, study design, blinding), the participants (inclusion and exclusion rules and number, including inclusion or exclusion of women with diabetes, either preexisting or gestational), the index test (range of wkGA when the scan was conducted, the EFW equation employed, and the threshold for screen positive), reference standard (outcome, wkGA at delivery, and the scan-to-delivery interval).

## Data extraction and synthesis

Sensitivity, specificity, positive and negative likelihood ratios (LRs) [9] were calculated from standard two-by-two tables, which had been extracted for each study by tabulating each of the

different definitions of screen positive with each of the different outcomes studied. The "hierarchical summary receiver–operating characteristics" (HSROC) model of Rutter and Gatsonis [10] was utilized for data synthesis. This method allows the results of studies to be combined despite variation between studies in the threshold employed for screen positive. The bivariate logit-normal (Reitsma) model [11] was used to calculate average estimates of sensitivity and specificity and respective variances, at a specific threshold, in analyses in which data were available from at least four studies. We also used meta-analysis to obtain a summary of the diagnostic odds ratios (DORs) [12]. Publication bias was assessed using the Deeks' funnel plot asymmetry test when data was available from a sufficient number of studies. Significant asymmetry was assumed at $P < 0.05$ [13]. Statistical analyses were performed using STATA version 14 (StataCorp LP, College Station, Texas), specifically, its METANDI, METAN, and MIDAS packages. Analysis and reporting was performed using the Preferred Reporting Items for Systematic Reviews and Meta-Analyses (PRISMA) guidelines (S1 PRISMA Checklist) [14].

## Results

### Study characteristics

Fig 1 is the literature search PRISMA flowchart. Out of 9,811 unique titles and 72 full paper reviews, we identified 41 studies [15–55] fulfilling the inclusion criteria, including a total of 112,034 participants. The study characteristics are presented in S1 Table. Six studies [18,27,33, 36,37,52] ($N = 53,935$) included unselected pregnancies, nine [23,29,31–33,35,43,45,53,54] ($N = 6,436$) were confined to low-risk pregnancies, and 26 [15–17, 9–22,24–26,28,30,34,38–42,44,46–51,55] ($N = 51,663$) recruited pregnancies at mixed risk. The list of the excluded studies and the reasons for the exclusion are presented in S2 Table.

### Quality assessment

The risk of bias, as assessed by the QUADAS-2 tool, is summarized in Fig 2 and presented in detail in S1 Fig. The Galvin 2017 study [29] was published as an abstract; hence, we used a different study from the same cohort (GENESIS study) [56] to assess the risk of bias. Two of the included studies [51,52] have been authored by some of the coauthors of this paper. We used the same criteria for the quality assessment and analysis. Only three studies—Sovio 2018 [52] (Pregnancy Outcome Prediction study), Galvin 2017 [29] (GENESIS study), and Peregrine 2007 [47]—blinded the results to the clinicians. Hence, the large majority of studies were at risk of bias in relation to the reference standard. The second most common risk of bias was in relation to flow and timing, as six studies [19,24,36,39,47,55] performed the ultrasound either prior to IOL or less than 72 hours before delivery, resulting in a very short interval between the scan and delivery. Conversely, two studies [18,27] had a very long interval (ultrasound <33 wkGA). Two studies [17,20] did not present data on the gestational age at delivery. Finally, three studies [23,48,54] were confined to pregnancies progressing beyond 41 wkGA and were classified as having "high applicability concerns due to patient selection".

### Meta-analysis results

Full details of the summary diagnostic performance are presented in Table 1. In summary, both definitions of ultrasonically suspected macrosomia (i.e., either EFW >4,000 g or >90th percentile) had >50% sensitivity for predicting LGA at birth. Many associations were similar regardless of the formula employed, but the positive LRs for the Hadlock formulae (ranging between 7.5 and 12) tended to be higher than for the Shepard formula (around 5). The

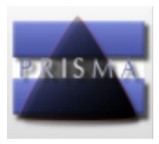

# PRISMA 2009 Flow Diagram

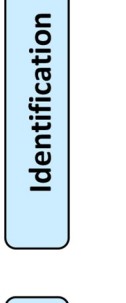

Records identified through
database searching
(n = 11,551)

Additional records identified
through other sources
(n = 2)

Records after duplicates removed
(n = 9957)

Records screened
(n = 268)

Records excluded
(n = 196)

Full-text articles assessed
for eligibility
(n = 72)

Full-text articles excluded
(n = 31)
High risk/ intrapartum
only (n=15)
Reviews/abstracts (n=4)
No DTA studies / unable
to construct 2x2 tables
(n=8)
Case-control (n=4)

Studies included in
qualitative synthesis
(n = 41)

Studies included in
quantitative synthesis
(meta-analysis)
(n = 41)

**Fig 1. PRISMA flow diagram.**

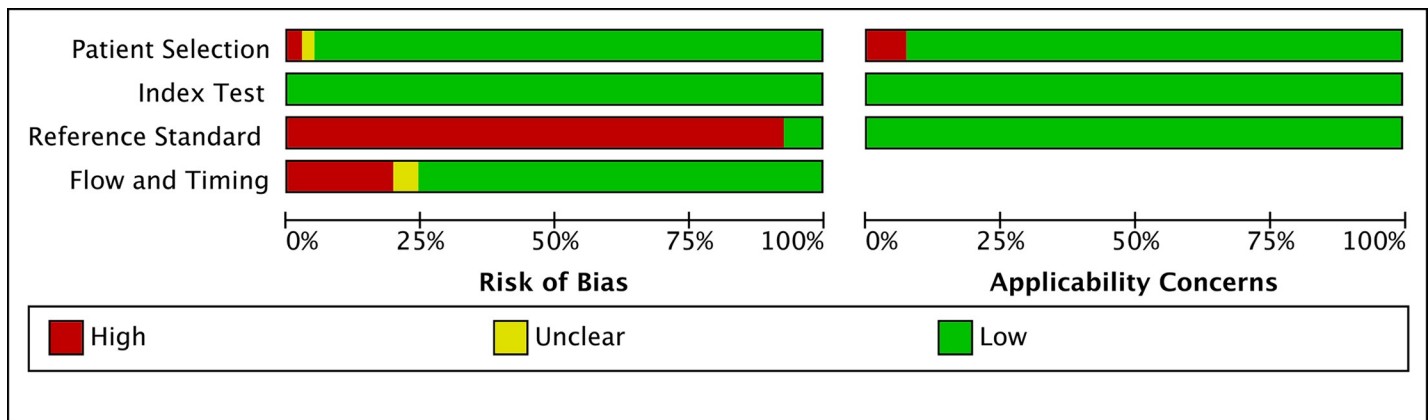

**Fig 2. Summary of bias assessment using the QUADAS-2 tool of the studies included in the meta-analysis.** QUADAS 2, Quality Assessment of Diagnostic Accuracy Studies.

**Table 1. Summary diagnostic performance of suspected LGA to predict adverse perinatal outcome.**

| Diagnostic test | Studies | Patients | Summary sensitivity | Summary specificity | Positive LR (95% CI) | Negative LR (95% CI) |
|---|---|---|---|---|---|---|
| | | | (95% CI) | (95% CI) | | |
| **Outcome: Birthweight >4,000 g (or 90th centile)** | | | | | | |
| EFW (any) >4,000 g (or 90th centile) | 30 | 80,045 | 53.2% | 93.9% | 8.74 | 0.50 |
| | | | (47.2%–59.1%) | (91.9%–95.5%) | (6.84–11.17) | (0.44–0.56) |
| EFW (Hadlock-AC/FL/HC/BPD) | 9 | 22,073 | 63.1% | 94.3% | 11.13 | 0.39 |
| | | | (49.1%–75.2%) | (90.9%–96.5%) | (8.24–15.04) | (0.28–0.55) |
| EFW (Hadlock- AC/FL/BPD) | 10 | 17,110 | 55.1% | 92.9% | 7.77 | 0.48 |
| | | | (44.1%–65.7%) | (89.7%–95.2%) | (5.55–10.89) | (0.38–0.61) |
| EFW (Hadlock- AC/FL/HC) | 7 | 60,648 | 55.2% | 94.9% | 11.84 | 0.47 |
| | | | (45.7%–64.2%) | (92.4%–96.6%) | (7.46–15.74) | (0.39–0.58) |
| EFW (Hadlock- AC/FL) | 9 | 16,736 | 60.5% | 92.0% | 7.54 | 0.43 |
| | | | (50.7%–69.5%) | (89.4%–93.7%) | (6.13–9.29) | (0.34–0.54) |
| EFW (Hadlock- AC/BPD) | 6 | 13,617 | 62.9% | 93.7% | 9.99 | 0.40 |
| | | | (36.1%–83.5%) | (85.9%–97.3%) | (6.40–15.58) | (0.21–0.75) |
| EFW (Shepard) | 7 | 14,060 | 73.7% | 85.1% | 4.96 | 0.31 |
| | | | (54.4%–86.9%) | (76.5%–90.9%) | (3.29–7.48) | (0.17–0.56) |
| AC >36cm (or 90th centile) | 5 | 10,543 | 57.8% | 92.3% | 7.56 | 0.46 |
| | | | (39.6%–74.2%) | (88.7%–94.9%) | (5.85–9.77) | (0.30–0.68) |
| **Outcome: Birthweight >4,500 g (or 97th centile)** | | | | | | |
| EFW (any) >4,000 g (or 90th centile) | 5 | 51,686 | 67.5% | 89.7% | 6.58 | 0.36 |
| | | | (47.8%–82.6%) | (79.1%–95.3%) | (2.78–15.58) | (0.20–0.65) |
| **Outcome: Shoulder dystocia** | | | | | | |
| EFW (any) >4,000 g (or 90th centile) | 6 | 26,264 | 22.0% | 89.6% | 2.12 | 0.87 |
| | | | (9.9%–42.0%) | (80.8%–94.6%) | (1.34–3.35) | (0.74–1.02) |

**Abbreviations:** AC, abdominal circumference; BPD, Biparietal diameter; CI, confidence interval; EFW, estimated fetal weight; FL, femur length; HC, head circumference; LR, likelihood ratio

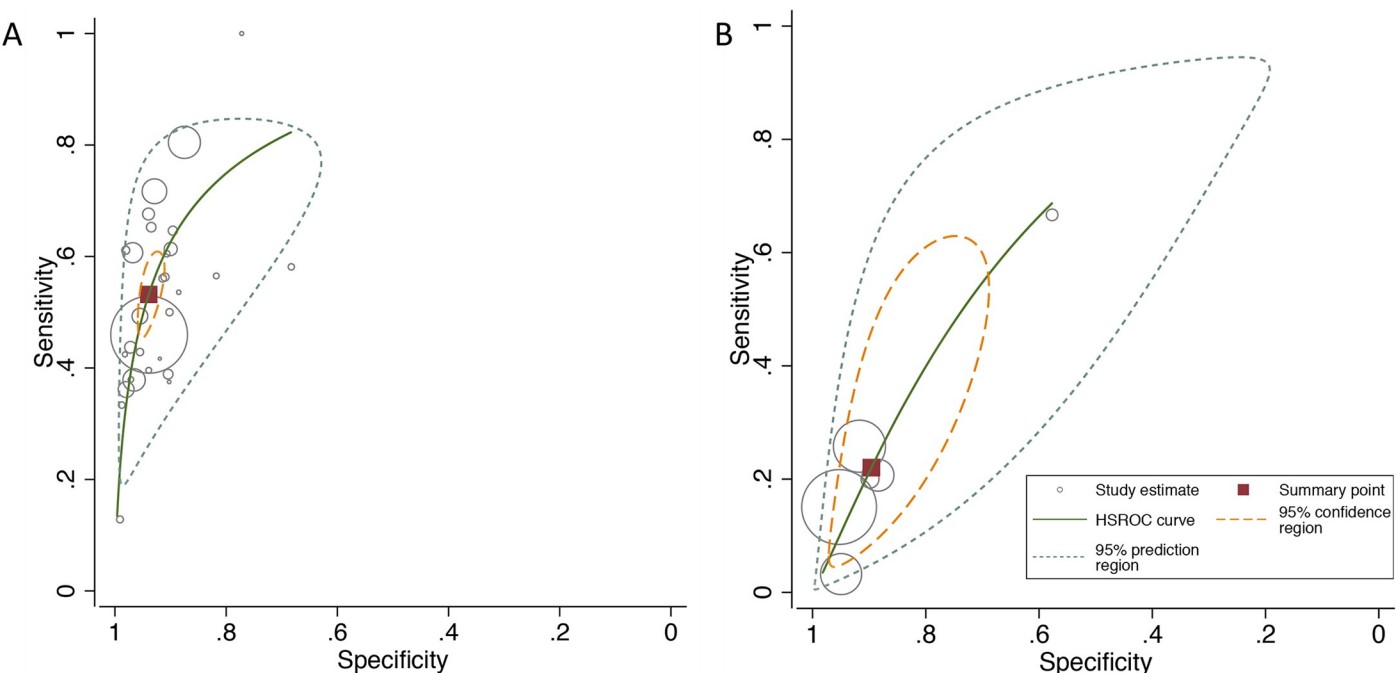

**Fig 3.** Summary ROC curves for the diagnostic performance of EFW >4,000 g (or 90th centile) at predicting (A) macrosomia at birth (birthweight above 4,000 g or above the 90th centile) and (B) shoulder dystocia. EFW, estimated fetal weight.

performance of definitions using just the AC was similar to using an ultrasonic EFW. The sensitivity for predicting severe macrosomia at birth of suspected LGA was around 70%. However, macrosomia (EFW >4,000 g or >90th centile) had a lower (22%) sensitivity for predicting shoulder dystocia, although the association was statistically significant and the positive LR was approximately 2.

Fig 3 has summary ROC curves for shoulder dystocia and macrosomia. For the prediction of macrosomia at birth, most of the large studies were close to the point estimate, and only a few small studies were outside the prediction intervals. For shoulder dystocia, most studies reported sensitivities below 30%, and only one study [55] reported a sensitivity of >50%. However, in this study, the total number of shoulder dystocia cases was very small ($n = 3$). Fig 4 and Fig 5 present graphs of the pooling of DORs for macrosomia and shoulder dystocia, respectively. There was significant heterogeneity for the prediction of macrosomia but not for the prediction of shoulder dystocia.

Only three studies—Crimmins 2018 [25], Galvin 2017 [29], and Sovio 2018 [52]—reported neonatal unit admission as an outcome, and a meta-analysis was not feasible. However, none of the studies reported statistically significant results with positive LRs of 0.73 (95% confidence interval [CI] 0.36–1.48), 1.39 (95% CI 0.97–2.00), and 1.33 (95% CI 0.80–2.22), respectively. Only the Sovio 2018 [52] study reported on 5-minute Apgar score of less than 7 and neonatal metabolic acidosis with positive LRs of 1.94 (95% CI 0.66–5.75) and 1.08 (95% CI 0.28–4.18), respectively. Moreover, the Sovio 2018 study was the only one that reported on neonatal hypoglycaemia and neonatal jaundice with positive LRs of 1.9 (95% CI 1.1–3.4) and 1.2 (95% CI 0.6–2.4), respectively.

The analysis demonstrated no significant evidence of publication bias ($P = 0.57$) when evaluated using Deeks' funnel plot asymmetry test (S2 Fig).

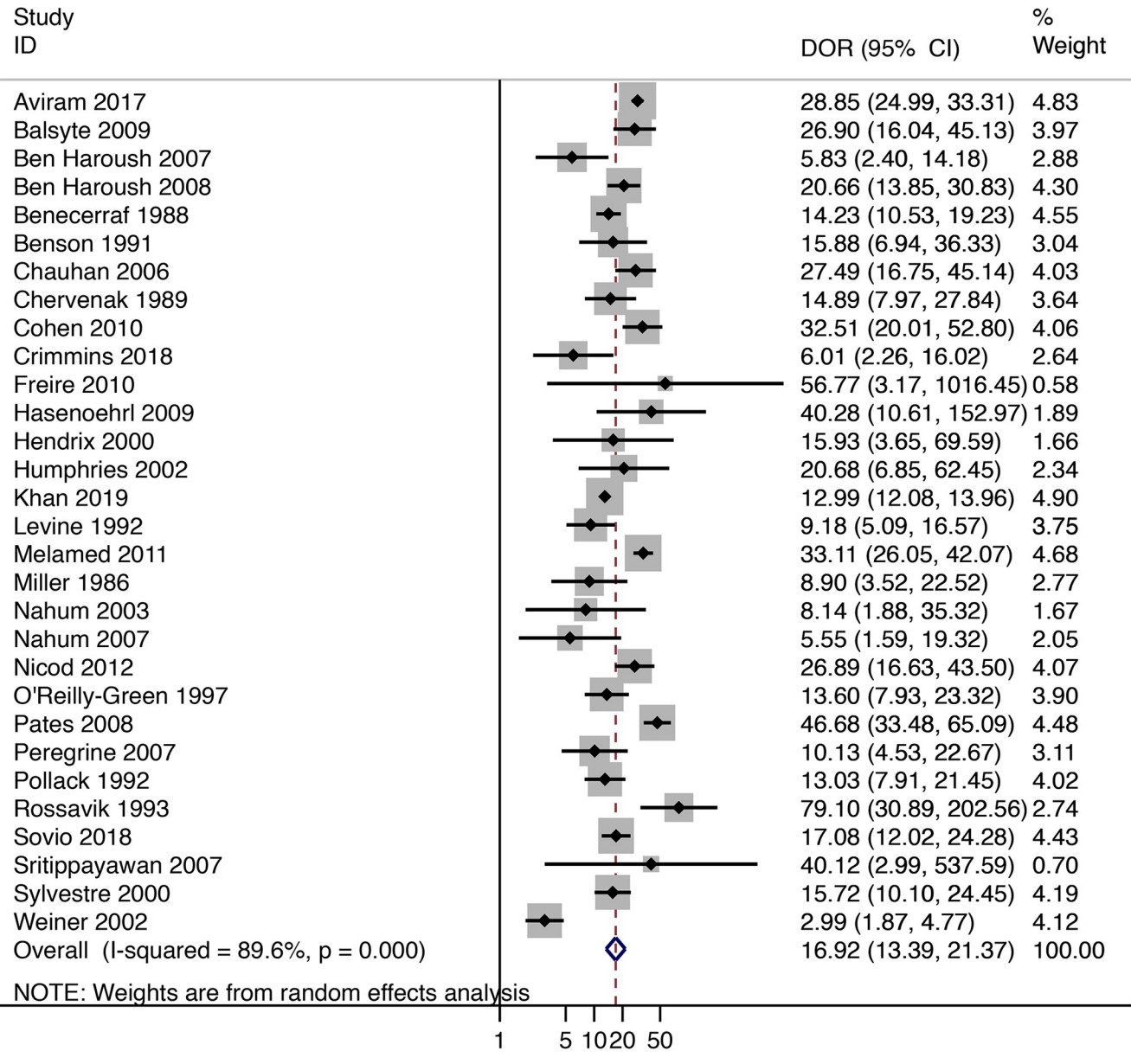

**Fig 4. Diagnostic performance of EFW >4,000 g (or 90th centile) at predicting macrosomia at birth (birthweight above 4,000 g or above the 90th centile).** EFW, estimated fetal weight.

## Discussion

The main conclusion of this analysis is that an ultrasonic EFW indicating an increased risk of a large baby was strongly associated with delivering a macrosomic infant, but it was only weakly associated with the risk of shoulder dystocia. When the EFW was calculated using the widely employed Hadlock method, the positive LRs for macrosomia were in the region of 7 to 12, whereas they were approximately 2 in relation to the risk of shoulder dystocia.

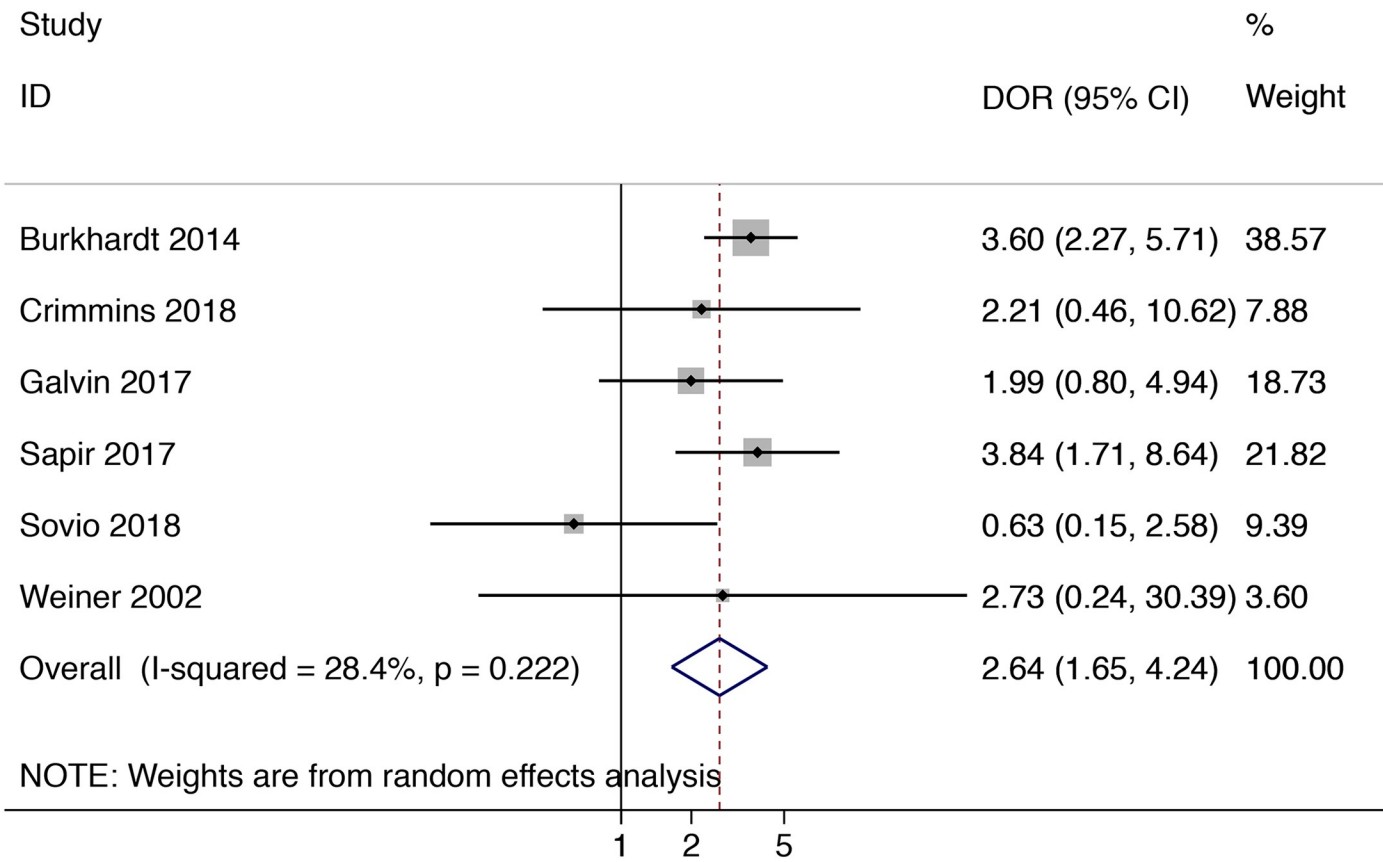

**Fig 5. Diagnostic performance of EFW >4,000 g (or 90th centile) at predicting shoulder dystocia.** EFW, estimated fetal weight.

This is the largest systematic review on the prediction of macrosomia and the only study that was focused on low- and mixed-risk populations from the perspective of using third-trimester ultrasound as routine screening in all pregnancies. We reported on multiple ultrasound markers and formulas. Moreover, we also reported on the prediction of shoulder dystocia, which is a major perinatal complication, the prevention of which would be a major aim of the routine ultrasound screening. The main limitation of this study is that there was significant heterogeneity between the studies in the ability to predict a macrosomic infant, as the forest plot of DORs indicates. The source of this heterogeneity is unclear, but it could relate to differences in the quality of the performance of the diagnostic test, such as the quality of the imaging equipment, the skill and training of sonographers, and the characteristics of the population. Finally, despite the large amount of studies included, only three studies [25, 29, 52] reported any outcomes of neonatal morbidity, and a meta-analysis was not feasible.

In the current study, we incorporated previously published data from the POP study (Sovio 2018) [52], which included nulliparous women who had a research scan at 36 wkGA, which was blinded in most cases to the clinicians. We found that the DOR (95% CI) from the POP study was very similar to the summary DOR derived from all of the other studies, which suggests that the results from the POP study are likely to be generalizable. The POP study was one of only a few identified that blinded the ultrasound result. Another blinded study, conducted in seven centers across Ireland between 2012 and 2015, the GENESIS study (Galvin 2017) [29], was a prospective cohort study of 2,772 nulliparous pregnant women. The results of the GENESIS study have only

been published in conference proceedings [29] and include the outcome of shoulder dystocia but not macrosomia. Interestingly, neither the POP study nor the GENESIS study observed a statistically significant association between ultrasonic LGA and shoulder dystocia. When blinded and unblinded studies were combined, the meta-analysis demonstrated that ultrasound may be predictive of shoulder dystocia, albeit weakly. However, the associations observed in the other studies may be due to ascertainment bias. Specifically, if the fetus is suspected to be large on the basis of the EFW, the staff attending the delivery may have a lower threshold for using maneuvers for shoulder dystocia in the event of any delay. They may also be more likely to document a given delay as being due to shoulder dystocia. Hence, unblinded studies could result in stronger associations with shoulder dystocia through ascertainment bias. The fact that ultrasonic EFW is relatively poor as a predictor of shoulder dystocia is not unexpected, given that the actual birth weight of the baby is also not strongly predictive of the outcome: the majority of cases of shoulder dystocia involve a normal birth weight infant [57].

Finally, ultrasonic suspicion of a large baby is a clinical situation where there is evidence that knowledge of the scan result may itself cause complications. Multiple studies have demonstrated that women who have a false positive diagnosis of fetal macrosomia based on EFW are more likely to be delivered by emergency caesarean section [58,59]. This finding underlines the potential for harm caused by screening low-risk women. Research studies in which the results of the scan are revealed could lead to associations with adverse outcomes that were caused by an iatrogenic harm from a false positive result. Conversely, analysis of studies in which the scan was revealed may fail to show true associations with adverse outcome as knowledge of the scan result led to interventions that mitigated the risk.

We conclude that ultrasonically suspected LGA in the general population has quite good diagnostic effectiveness for macrosomic birth weight. However, it is not strongly predictive of the risk of associated complications, such as shoulder dystocia. Similar observations have been made in relation to ultrasonically suspected SGA [60, 61]. That study indicated that reduced fetal abdominal growth velocity helped discriminate between healthy SGA babies and those that were at increased risk of complications. Interestingly, the analogous finding is also true in LGA babies, in whom the combination of LGA and accelerated abdominal growth velocity was associated with the risk of neonatal morbidity [52]. We believe that future studies should address the other factors which help differentiate those suspected LGA fetuses which are at the greatest risk of complications.

## Supporting information

**S1 PRISMA Checklist. PRISMA checklist.**
(DOC)

**S1 Text. Literature search strategy for Medline and EMBASE (from inception to May 2020).**
(DOCX)

**S1 Table. Characteristics of the studies included in the meta-analysis.**
(DOCX)

**S2 Table. List of studies excluded from the meta-analysis and reason for exclusion.**
(DOCX)

**S1 Fig. Risk of bias assessment using the QUADAS 2 tool.** QUADAS 2, Quality Assessment of Diagnostic Accuracy Studies
(TIFF)

**S2 Fig. Deeks' funnel plot asymmetry test.**
(TIFF)

## Acknowledgments

The views expressed here are those of the authors and not necessarily those of the NHS, the NIHR, or the Department of Health.

## Author Contributions

**Conceptualization:** Alexandros A. Moraitis, Peter Brocklehurst, Alexander E. P. Heazell, Jim G. Thornton, Stephen C. Robson, Aris Papageorghiou, Gordon C. Smith.

**Data curation:** Alexandros A. Moraitis, Norman Shreeve, Ulla Sovio.

**Formal analysis:** Alexandros A. Moraitis, Norman Shreeve, Ulla Sovio, Gordon C. Smith.

**Funding acquisition:** Peter Brocklehurst, Jim G. Thornton, Stephen C. Robson, Aris Papageorghiou, Gordon C. Smith.

**Investigation:** Alexandros A. Moraitis.

**Methodology:** Alexandros A. Moraitis, Ulla Sovio, Peter Brocklehurst, Gordon C. Smith.

**Supervision:** Gordon C. Smith.

**Visualization:** Alexandros A. Moraitis.

**Writing – original draft:** Alexandros A. Moraitis, Gordon C. Smith.

**Writing – review & editing:** Alexandros A. Moraitis, Norman Shreeve, Ulla Sovio, Peter Brocklehurst, Alexander E. P. Heazell, Jim G. Thornton, Stephen C. Robson, Aris Papageorghiou, Gordon C. Smith.

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
