## [Editor Report · Decision Letter 0]

15 Jan 2020

Dear Dr Smith, 

Thank you for submitting your manuscript entitled "Universal third trimester ultrasonic screening using fetal macrosomia in the prediction of adverse perinatal outcome, a systematic review and meta-analysis of diagnostic test accuracy." for consideration by PLOS Medicine.

Your manuscript has now been evaluated by the PLOS Medicine editorial staff [as well as by an academic editor with relevant expertise] and I am writing to let you know that we would like to send your submission out for external peer review.

**Please be aware that, due to the voluntary nature of our reviewers and academic editors, manuscript assessment may be subject to delays during the holiday season. Thank you for your patience.**

Kind regards,

Helen Howard, for Clare Stone PhD

Acting Editor-in-Chief

PLOS Medicine

plosmedicine.org

---

## [Decision Letter · Decision Letter 1]

9 Apr 2020

Dear Gordon,

Thank you very much for submitting your manuscript "Universal third trimester ultrasonic screening using fetal macrosomia in the prediction of adverse perinatal outcome, a systematic review and meta-analysis of diagnostic test accuracy." (PMEDICINE-D-20-00081R1) for consideration at PLOS Medicine. Please accept my sincere apologies for the unusual delay in getting back to you about it. One of the reviewers was very late submitting. 

[LINK]

In light of these reviews, I am afraid that we will not be able to accept the manuscript for publication in the journal in its current form, but we would like to consider a revised version that addresses the reviewers' and editors' comments. Obviously we cannot make any decision about publication until we have seen the revised manuscript and your response, and we plan to seek re-review by one or more of the reviewers. 

We expect to receive your revised manuscript by Apr 30 2020 11:59PM. Please email us (plosmedicine@plos.org) if you have any questions or concerns.

We look forward to receiving your revised manuscript. 

Sincerely,

Clare Stone, PhD

Managing Editor 

PLOS Medicine

plosmedicine.org

Title – Please insert a colon before the study descriptor in Universal third trimester ultrasonic screening using fetal macrosomia in the prediction of adverse perinatal outcome, a systematic review and meta-analysis of diagnostic test accuracy.

Abstract – please use p values, where possible, where 95%Cis are used (also elsewhere, as needed) and please add a sentence on the study’s limitations as the final sentence of the abstract

Thank you for providing the PRISMA checklist – please use sections and paragraphs instead of page numbers as these can change on formatting and revision. 

Comments from the reviewers:

Reviewer #1: The study question (3rd trimester US screening for macrosomia) is a well-known, but not evidence based answered one. It is an important problem to the community of clinical obstetricians in the high- and middle-income countries.

The design and the mothodology of the study is well organized. The results are clearly written and the consequnezes for the day-to-day work of obstetrical clinicians and for future scientific study questions are mentioned.

The manuscript can be published without revision, may I ask for correction of some typing errors (like in the short running title marcosomia - macrosomia!).

Reviewer #2: See attachment

Michael Dewey

Reviewer #3: Overall: a well written paper: clear and concise, easy to read, clinically relevant.

Abstract: In the abstract there is a reference to ultrasound screening and breech. Why, this is not incorporated in the introduction, nor is a reference given. Seems a bit out of place.

The aim is clear and well defined. Moreover, the authors address an important clinical problem: can shoulder dystocia be predicted?.

I have only one concern. I miss information on the inclusion of just abstracts (i.e. non-peer reviewed studies) in this systematic review and meta-analysis (since the abstract of Galvin 2017 is included) . Did the authors check if study protocols were published in a registry or were authors contacted in order to assess the risk of bias? If authors are unable to share their protocol, their study should not included. Please add.

[LINK]

---

## [Decision Letter · Decision Letter 2]

27 May 2020

Dear Dr. Smith,

Thank you very much for re-submitting your manuscript "Universal third trimester ultrasonic screening using fetal macrosomia in the prediction of adverse perinatal outcome: a systematic review and meta-analysis of diagnostic test accuracy." (PMEDICINE-D-20-00081R2) for review by PLOS Medicine.

I have discussed the paper with my colleagues and the academic editor and it was also seen again by xxx reviewers. I am pleased to say that provided the remaining editorial and production issues are dealt with we are planning to accept the paper for publication in the journal.

[LINK]

We look forward to receiving the revised manuscript by Jun 03 2020 11:59PM. 

Sincerely,

Adya Misra, PhD

Senior Editor 

PLOS Medicine

plosmedicine.org

Requests from Editors:

Please update your literature search to the end of April to ensure no recent publications have been missed. 

Abstract

Please add a sentence, say, to the abstract to summarize the included studies - for example, quoting the range of study dates and categories of study design (e.g., X RCTs, Y cohort studies ...). 

Please add a new final sentence to the "methods and findings" subsection of your abstract, to quote 2-3 of the study's main limitations. 

In your abstract and throughout the paper, please add p values alongside 95% CI, where available

Author summary 

Please rephrase level 1 evidence here as it may not be accessible to all readers. You may wish to include this as a second bullet point 

Please rephrase lines 90-96 using more accessible language 

Please break the first subsection of the "author summary" down into 2-3 individual points. 

Introduction

Please can you add a space between text and reference brackets

References

Please format the bibliography in Vancouver style

Competing Interests- please add a sentence to note GS is an Academic Editor at PLOS Medicine

At line 70, please begin the sentence, "In this study, we found that ..." or similar. 

Please refer to the attached PRISMA checklist in the methods section of your main text. 

To the discussion section of your main text, please add a concise discussion of study strengths and limitations. 

Please remove the funding and competing interest information from the end of the text (this will appear in the metadata in the event of publication, via the submission form). 

Please avoid "P=0.000" in the figures. 

Comments from Reviewers:

Reviewer #1: The revised paper is ready for publication.

Reviewer #2: The authors have addressed my points.

Michael Dewey

[LINK]

---

## [Editor Report · Decision Letter 3]

9 Sep 2020

Dear Prof. Smith, 

On behalf of my colleagues and the academic editor, Dr. Eva Pajkrt, I am delighted to inform you that your manuscript entitled "Universal third trimester ultrasonic screening using fetal macrosomia in the prediction of adverse perinatal outcome: a systematic review and meta-analysis of diagnostic test accuracy." (PMEDICINE-D-20-00081R3) has been accepted for publication in PLOS Medicine. 

PRODUCTION PROCESS

PRESS

PROFILE INFORMATION

Thank you again for submitting the manuscript to PLOS Medicine. We look forward to publishing it. 

Best wishes, 

Adya Misra, PhD

Senior Editor 

PLOS Medicine

plosmedicine.org